# Microbiologic Findings in a Cohort of Patients with Erythema Migrans

**DOI:** 10.3390/microorganisms12010185

**Published:** 2024-01-17

**Authors:** Eva Ružić-Sabljić, Vera Maraspin, Petra Bogovič, Tereza Rojko, Katarina Ogrinc, Martina Jaklič, Franc Strle

**Affiliations:** 1Institute of Microbiology and Immunology, Medical Faculty, University of Ljubljana, 1000 Ljubljana, Slovenia; 2Department of Infectious Diseases, University Medical Centre Ljubljana, 1525 Ljubljana, Slovenia; vera.maraspin@kclj.si (V.M.); petra.bogovic@kclj.si (P.B.); tereza.rojko@kclj.si (T.R.); katarina.ogrinc@kclj.si (K.O.); franc.strle@kclj.si (F.S.); 3Centre for Clinical Research, University Medical Centre Ljubljana, 1000 Ljubljana, Slovenia; martina.jaklic@kclj.si

**Keywords:** erythema migrans, Lyme borreliosis, *Borrelia afzelii*, culture, serology

## Abstract

Erythema migrans (EM) is the initial and the most frequent clinical manifestation of Lyme borreliosis (LB). Herein, we report on the capacity of culture and serology for the demonstration of *Borrelia* infection in a cohort of 292 patients diagnosed with typical EM at a single medical center. The median duration of EM at diagnosis was 12 days, and the largest diameter was 16 cm; 252 (86.3%) patients presented with solitary EM, whereas 40 (13.7%) had multiple EM. A total of 95/292 (32.5%) patients had positive IgM, and 169 (57.9%) had positive IgG serum antibodies; the *Borrelia* isolation rate was 182/292 (62.3%). The most frequent species by far was *B. afzelii* (142/148, 95.9%) while *B. garinii* (2.7%) and *B. burgdorferi* s.s. (1.4%) were rare. IgM seropositivity was associated with a younger age, multiple EM and the absence of underlying chronic illness; IgG seropositivity was associated with the duration of EM at diagnosis, the diameter of the EM, having had a previous episode of LB and the absence of symptoms at the site of the EM. Furthermore, the *Borrelia* isolation rate was statistically significantly lower in patients with positive *Borrelia* IgM antibodies. Although microbiologic analyses are not needed for the diagnosis of typical EM, they enable insights into the etiology and dynamics of the immune response in the course of early LB.

## 1. Introduction

Lyme borreliosis (LB) is the most common tick-borne disease in the Northern Hemisphere. It is caused by several species of *Borrelia burgdorferi* sensu lato complex (Lyme *Borrelia*). All *Borrelia* species use vertebrates as their reservoir hosts and ticks of the Ixodidae complex as vectors for transmission [1], but only some of them are associated with disease in humans. In Europe, LB is typically caused by *Borrelia afzelii*, *Borrelia garinii*, *Borrelia bavariensis*, *Borrelia burgdorferi* sensu stricto and *Borrelia spielmanii,* while in North America, *B. burgdorferi* sensu stricto is a highly predominant etiologic agent of the disease in humans [2,3].

The first sign of LB is usually an expanding skin lesion, erythema migrans (EM), that appears at the site of an infected tick bite within days to weeks after the bite [3]. However, since tick bites are typically painless and may occur at body sites that are difficult to perceive, several patients do not recall a tick bite. Nevertheless, if untreated, *Borrelia* may disseminate from the skin to other organ systems, including the nervous system, joints, or heart [2,3].

Erythema migrans is the most common clinical manifestation of LB. It is the only clinical manifestation that is sufficiently distinctive to allow a clinical diagnosis in the absence of a positive laboratory test. The three essential bases for a diagnosis of EM are (i) compatible information on the course of the skin lesion; (ii) recognition of the characteristic appearance; and (iii) an epidemiologic link (a person who lives in or has recently traveled to areas endemic for LB) [4]. Microbiologic findings are essential for the diagnosis of atypical EM skin lesions and all other manifestations of LB. However, in patients with typical EM, microbiological diagnosis is important for the research of the etiology of the skin disorder, as well as for monitoring the development and dynamics of the immune response in the course of early disease [2,3]. Herein, we report on the assessment of the capacity of culture and serology for demonstrations of *Borrelia* infection in a cohort of patients diagnosed with typical EM at a single medical center.

## 2. Materials and Methods

### 2.1. Patients

All patients with typical EM, diagnosed in 2013 at the LB Outpatient Clinic of the Department of Infectious Diseases, University Medical Centre Ljubljana, Slovenia, who had not received antibiotics before the examination; who were tested for the presence of *Borrelia* antibodies in serum; who had a skin biopsy and culture of the specimen for the presence of *Borrelia* performed; and who were examined by the authors of this paper, were included in this study.

Epidemiologic and clinical data were obtained prospectively using a questionnaire.

### 2.2. Definitions

EM was defined as an expanding erythematous skin lesion, with or without central clearing, that developed days to weeks after a tick bite or exposure to ticks and had a diameter ≥ 5 cm. If <5 cm in diameter, a history of a tick bite, a delay in the appearance of at least 2 days and an expanding rash at the bite site were required for a diagnosis of EM. Multiple EM was defined as the presence of ≥2 skin lesions, at least one of which fulfilled the size criteria (≥5 cm) for solitary EM [5].

### 2.3. Microbiological Analyses

To confirm *Borrelia* infection, all patients were tested for the presence of specific IgM and IgG antibodies in serum as well as for the presence of *Borrelia* in a skin biopsy specimen. All microbiological analyses were performed at the Institute of Microbiology and Immunology, Medical Faculty, University of Ljubljana.

#### 2.3.1. Serological Evaluation

Specific IgM and IgG antibodies were examined in blood samples via a commercially available chemiluminescence (CLIA) test produced by DiaSorin, Italy. Recombinant antigens OspC and VlsE were included in the test to capture specific IgM, and VlsE only was used to capture specific IgG antibodies. The results were reported as qualitative and quantitative (antibody units per mL, AU/mL) according to the manufacturer’s recommendation [6].

#### 2.3.2. Borrelia Cultivation and Typing

To test for the presence of *Borrelia* in skin, a 3 mm punch biopsy was obtained from the expanding edge of the EM after local anesthesia, as reported elsewhere [7]. The skin biopsy specimen was immediately inoculated into a 10 mL tube with a modified Kelly-Pettenkofer (MKP) medium. The tubes were incubated at 33 °C for 9 weeks, weekly subcultured and checked for live spirochetes under dark-field microscopy [8]. Isolated *Borrelia* strains were characterized at the species level based on MluI-large restriction fragment polymorphism (MluI-LRFP) [9].

### 2.4. Statistical Analysis

Continuous variables were summarized using median values and interquartile ranges (IQRs), and discrete variables were summarized using counts and percentages. The Shapiro–Wilk test was used to examine the normality assumption of the continuous variables. Differences between groups were analyzed with independent-sample *t*-tests for continuous variables, and Fisher’s exact tests or Pearson’s Chi-square test were used for categorical variables.

To assess the association of the presence of microbiologic parameters (presence of *Borrelia* IgM antibodies in serum, presence of *Borrelia* IgG antibodies in serum, isolation of *Borrelia* from skin) and several clinical parameters (sex, age, underlying chronic illness, previous LB, tick bite, duration of EM at diagnosis, diameter of EM at diagnosis, multiple EM, symptoms at the site of EM, constitutional symptoms), we estimated three separate logistic regression models with individual microbiologic parameters representing dependent variables. *p* values < 0.05 were considered statistically significant. The data were analyzed using SPSS version 25.0 (SPSS Inc., Chicago, IL, USA).

## 3. Results

In 2013, 520 adult patients with typical EM were diagnosed at our LB Outpatient Clinic, Ljubljana, Slovenia. Of these, 292 (141 females and 151 males) fulfilled all inclusion criteria (had not received antibiotics before the examination, were tested for the presence of *Borrelia* antibodies in serum, had a skin biopsy and culture of the skin specimen for the presence of *Borrelia* performed, and were examined by the authors of this paper) and were enrolled in the present study. The basic clinical findings for these 292 patients are depicted in Table 1. All 292 patients had typical EM; 144 (49.3%) of them recalled a tick bite at the site of a later EM. The median duration of EM at diagnosis was 12 days, and the largest diameter of the lesion was 16 cm. Of the 292 patients, 252 (86.3%) had solitary EM, while in 40 (13.7%) patients, more than one skin lesion was present. Nearly half of the patients reported symptoms at the site of the EM (most often mild itching, rarely local burning or pain); approximately one third had constitutional symptoms such as fatigue, headache, myalgia and/or arthralgia that newly appeared or intensified with the onset of the EM skin lesion.

Microbiologic findings are shown in Table 2. Of the 292 patients, 95 (32.5%) had positive serum IgM antibodies; in 23 (7.9%), the result was borderline, while 174 (59.6%) patients tested negative. The corresponding findings for IgG antibodies were 169 (57.9%), 25 (8.6%) and 98 (33.6%), for positive, borderline and negative results, respectively. Of the 292 patients, 195 (66.8%) were IgM- and/or IgG-seropositive.

We isolated *Borrelia* from the EM skin lesion in 182/292 (62.3%) patients. *Borrelia* growth was apparent 11 to 58 (median 24) days after skin biopsy. The distribution of days from skin biopsy to the detection of growth in culture is shown in Figure 1.

Typing was performed for the majority of isolates (148/182, 81.32%). The far most frequent species was *B. afzelii* (142/148, 95.9%), while *B. garinii* and *B. burgdorferi* s.s. were rare (4/148, 2.7% and 2/148, 1.4%, respectively).

As shown in Table 3, IgM seropositivity was associated with a younger age (median age of IgM-positive patients was 44 years vs. 53 years in IgM-negative/borderline patients; *p* < 0.001), while IgG seropositivity was positively associated with the duration of EM at diagnosis (median duration of EM in IgG-positive patients was 14 days vs. 9 days in IgG-negative/borderline patients; *p* = 0.001) and the diameter of the EM (median diameter of EM in IgG-positive patients was 19 cm vs. 12 cm in IgG-negative/borderline patients; *p* = 0.003).

Furthermore, patients with positive IgM had an underlying chronic illness less often than the IgM-negative/borderline patients (28/95, 25.9% vs. 80/187, 42.8%; *p* = 0.0410), but multiple EM was experienced more frequently than in IgM-negative/borderline patients (24/95, 25.3% vs. 16/187, 8.6%; *p* = 0.0003). Patients with positive IgG antibodies more often recalled having had a previous episode of LB (42/169, 24.8% vs. 15/123, 12.2%; *p* = 0.0109), and less often reported symptoms at the site of the EM skin lesion than those with negative/borderline IgG (66/169, 39.1% vs. 72/123, 58.5%; *p* = 0.0015). The only statistically significant finding associated with the skin culture results was that the *Borrelia* isolation rate was lower in patients with positive IgM than in those with negative/borderline findings (40/182, 22.5% vs. 55/110 (50%; *p* < 0.0001) (Table 4).

According to statistical models, age was statistically significantly associated with the presence of *Borrelia* IgM and IgG antibodies in serum; however, IgM seropositivity was associated with a younger age (Table 5), while IgG seropositivity was associated with older age (Table 6). Furthermore, *Borrelia* IgM seropositivity was statistically significantly associated with multiple EM (Table 5), while the presence of *Borrelia* IgG antibodies in the serum was positively associated with previous LB, the duration of EM at diagnosis and the diameter of EM at diagnosis (Table 6). Among the parameters tested for their association with the isolation of *Borrelia* from skin, a statistically significant association was found only for underlying chronic illness (Table 7).

## 4. Discussion

Our study encompassed 292 adult Slovenian patients with EM, an early and by far the most frequent manifestation of LB. None of these patients had received antibiotics before the examination; all were tested for the presence of *Borrelia* antibodies in their sera, had a skin biopsy and culture of the specimen for the presence of *Borrelia* performed and were examined by the authors of this paper. The fact that clinical and epidemiological information on these patients was obtained prospectively using a questionnaire, that all patients had a typical EM and that each of them fulfilled all pre-defined inclusion criteria enabled a rather homogeneous and large study group, well-suited for the assessment of microbiological characteristics of early LB. These characteristics had been quite extensively studied in the first two decades after the discovery of Lyme disease in the USA, but not so much during the last few decades, i.e., using more recent serological approaches. In addition, both serology and *Borrelia* culturing in the same patients, and the typing of the obtained isolates, were rarely performed.

Of the 292 patients, 95 (32.5%) had positive IgM *Borrelia* antibodies in sera and 169 (57.9%) had positive *Borrelia* IgG antibodies; overall, 195 (66.8%) patients were IgM- and/or IgG-positive. Thus, one third of our patients with typical EM did not have a serum *Borrelia* antibody response in a positive range. This is not a surprise, since it is well-known that early in the course of *Borrelia* infection, not all patients are seropositive [10,11]. However, it was rather unexpected that in early LB, with a median EM lesion duration of 12 days, the proportion of seropositivity was lower for IgM than for IgG antibodies. We do not have a reliable explanation for this finding; however, IgG seropositivity might be in part due to a previous *Borrelia* infection, either symptomatic (19.5% of our patients reported having had previous LB) or asymptomatic. The finding that, in our patients with an antecedent episode of LB, IgG antibodies were present in a higher proportion than in those without previous LB strengthens this hypothesis.

IgG seropositivity was associated with a longer duration and larger diameter of EM skin lesion at diagnosis, with information on having had a previous episode of LB and with a less frequent presence of symptoms at the site of the EM, while IgM seropositivity was associated with younger age, multiple EM and a less frequent presence of underlying illness. The associations of the duration of EM and the diameter of the skin lesion with IgG seropositivity, as well of that of multiple EM with IgM seropositivity, were predictable, while the other statistically significant associations were rather unexpected. All predictable associations were also found to be statistically significant in statistical models used in the present study, while the others—with the exception of the association of IgM seropositivity with younger age—were not.

In the present study, the *Borrelia* isolation rate from the skin of EM lesions was 62%, which is within the range of previously reported proportions (from 20% to almost 90%) [3,11,12]. The *Borrelia* recovery rate depends upon several factors. Since the number of *Borrelia* strains present in skin samples was found to be low, and the numbers of cultivable spirochetes and their different growth potential (which presents *Borrelia’s* natural make-up) in individual samples are rather unpredictable, the volume of specimen obtained is important [13,14,15]. Comparing quantitative PCR and culture, O’Rourke et al. confirmed that the number of *Borrelia* in culture-positive skin samples is significantly higher than in culture-negative specimens [13]. Wormser et al. enhanced the sensitivity of the culture by increasing the volume of material cultured [14]. However, the present study was not designed to assess the influence of sample volume on the *Borrelia* recovery rate from skin because the volume of the samples obtained with the 3 mm punch biopsy was approximately the same in all patients. Nevertheless, in the present study, several other clinical and laboratory parameters were tested for their potential association with the *Borrelia* skin culture results. The only statistically significant finding was that the isolation rate of *Borrelia* from skin was more than two times lower in patients with positive *Borrelia* IgM antibodies in sera than in patients with negative or borderline levels of IgM serum antibodies. However, among the parameters tested for their association with the isolation of *Borrelia* from skin in a statistical model, the association between *Borrelia* IgM seropositivity and positive *Borrelia* skin culture was not statistically significant. In the study of Strle and al. that compares the clinical characteristics of more than 1000 either culture-positive or -negative EM patients, the authors reported a correlation of culture positivity with older age, the location of the EM on extremities, a diameter of EM ≥ 5 cm and a time interval of >2 days between the tick bite and the EM’s appearance [16], while the only statistically significant association in our study was found for underlying chronic illness (Table 7). Some other studies also discuss the clinical features, laboratory findings and epidemiological data of erythema migrans patients [17,18,19].

*Borrelia* growth was detected on median day 24 after the inoculation of the skin specimen in the MKP culture medium. Since the medium was not checked for growth daily, but rather (usually) once per week, the detection time is most probably overestimated by a few days. *Borrelia* need time to adapt to artificial media and to multiply. It seems that the adaptation times are rather heterogeneous—in the present study, *Borrelia* growth was detected microscopically as early as day 11 and as late as day 58 after the inoculation of the skin specimen into the culture medium. Furthermore, it seems that the temporal occurrence does not follow a normal distribution, and that there are two main peaks: the majority of *Borrelia* strains multiplied sufficiently to be detected microscopically 21–27 days following the inoculation of the skin specimen into the culture medium, while several strains adapted as early as within the first 14 days (Figure 1). It is of interest that the two *B. burgdorferi* sensu stricto strains obtained in the present study were detected in culture rather late (4–5 weeks after inoculation), although strains of this species were reported to grow faster than strains of the other European *Borrelia* species that are pathogenic to humans [20]. Nevertheless, in addition to the period needed for adaptation, the time until culture positivity depends upon two main parameters: the number of *Borrelia* strains in the initial sample and the generation time deposited in the bacteria genome.

We do not have a reliable explanation for the negative correlation between the *Borrelia* isolation rate from the skin and the presence of *Borrelia* IgM (but not IgG) antibodies in serum (Table 4). However, low IgM seropositivity in EM patients with *Borrelia* isolation from skin and/or blood has been reported previously [21].

Our study revealed a pronounced predominance of *B. afzelii* among skin isolates (96%). This proportion was comparable or even slightly higher compared to most of the data published so far for Slovenia [9,13,22], possibly due to the fact that in the cohort of EM patients enrolled in the present study, those with clinical indications of neurologic involvement, which is often associated with *B. garinii*, were not included.

The strength of our study is its large and well-defined group of patients with typical EM for whom clinical and epidemiological information was obtained prospectively using a questionnaire and for whom we used exactly the same diagnostic approaches to confirm *B. burgdorferi* s. l. infection, including testing for the presence of *Borrelia* antibodies in sera and culturing the skin biopsy specimen for the presence of *Borrelia*. Our results, however, apply to adult patients but not necessarily to children, and to early LB caused by *B. afzelii* but not to other *Borrelia* species causing LB in humans.

## 5. Conclusions

In the group of 292 adult Slovenian patients with typical EM and median skin lesion duration of 12 days, 33% of patients had positive IgM and 58% had positive IgG antibodies, while the *Borrelia* isolation rate from the skin of the EM lesions was 62%. *Borrelia* growth was detected 11 to 58 (median 24) days after skin biopsy; 96% of the isolates were *B. afzelii*. IgM seropositivity was associated with younger age and multiple EM, while IgG seropositivity was associated with the duration of EM as well as the diameter of the EM at diagnosis, and with having had a previous episode of LB. The only statistically significant finding associated with the skin culture results was the lower *Borrelia* isolation rate in patients with positive serum IgM antibodies.

## Figures and Tables

**Figure 1 microorganisms-12-00185-f001:**
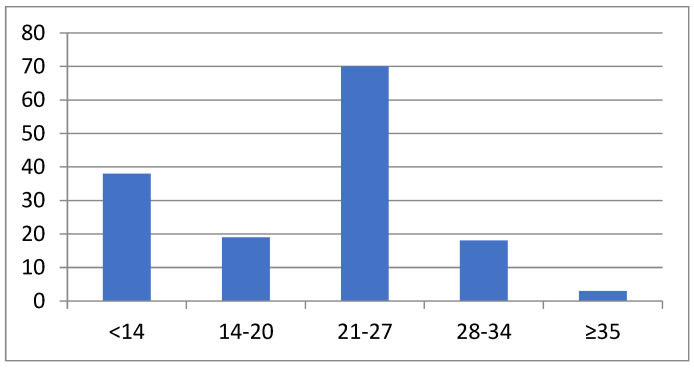
Days from skin biopsy to positive culture.

**Table 1 microorganisms-12-00185-t001:** Basic epidemiologic and clinical data on 292 patients with erythema migrans.

Female Sex	141 (48.3)
Age (years)	51 (40–60)
Underlying chronic illness	108 (37.0)
Previous LB	57 (19.5)
Tick bite *	144 (49.3)
Duration of EM at diagnosis	12 (4–29)
Diameter of EM at diagnosis	16 (10–24)
Multiple EM	40 (13.7)
Symptoms at the site of EM	138 (47.3)
Constitutional symptoms	99 (33.9)

Data are number (%) or median (IQR); LB, Lyme borreliosis; EM, erythema migrans; * at the site of the EM.

**Table 2 microorganisms-12-00185-t002:** Microbiologic findings in patients with erythema migrans at diagnosis.

***Borrelia* Antibodies in Serum**	
IgM-positive	95 (32.5)
borderline	23 (7.9)
negative	174 (59.6)
IgM levels: all patients	13.2 (7.3–32.4)
seropositive patients	54.3 (32.4–156.0)
IgG-positive	169 (57.9)
borderline	25 (8.6)
negative	98 (33.6)
IgG levels: all patients *	23.3 (7.4–75.0)
seropositive patients **	56.3 (28.0–147.6)
IgM- and/or IgG-positive	195 (66.8)
**Isolation of *Borrelia burgdorferi* sensu lato from skin**	
Isolation	182 (62.3%)
Time from skin biopsy to positive culture (days)	24 (13–26) range 11–58
*Borrelia* species ***	
*B. afzelii*	142 (95.9)
*B. garinii*	4 (2.7)
*B. burgdorferi* s.s.	2 (1.4)
IgM- and/or IgG- and/or *Borrelia* culture-positive	261 (89.4)

Data are number (%) or median (IQR); * information available for 286/292 patients; ** information available for 168/169 patients; *** of 182 isolates, 148 have been typed.

**Table 3 microorganisms-12-00185-t003:** Association of IgM seropositivity, IgG seropositivity and *Borrelia* skin culture positivity with patients’ age, duration of EM at diagnosis and diameter of EM at diagnosis.

Parameter	All Patients	IgM-pos	IgM-neg *	*p*	IgG-pos	IgG-neg *	*p*	Skin Culture-pos	Skin Culture-neg	*p*
Age (years)	51 (40–60)	44 (34–57)	53 (42–62)	**<0.001**	52 (40–61)	50 (38–59)	0.184	52 (40–61)	50 (38–60)	0.408
Duration of EM at diagnosis	12 (4–29)	11 (5–28)	12 (4–30)	0.900	14 (6–30)	9 (4–21)	**0.001**	9 (4–23)	14 (7–31)	0.095
Diameter of EM at diagnosis	16 (10–24)	18 (10–26)	15 (10–23)	0.606	19 (13–26)	12 (8–18)	**0.003**	15 (10–24)	17 (12–24)	0.853

EM, erythema migrans; pos, positive; neg *, negative or borderline; boldface is used for *p*-values < 0.05.

**Table 4 microorganisms-12-00185-t004:** Association of IgM seropositivity, IgG seropositivity and *Borrelia* skin culture positivity with discrete clinical and laboratory parameters.

	Number of Patients No = 292	IgM-pos N=95	IgM-neg * N=187	*p* Value	IgG-pos N=169	IgG-neg * N=123	*p* Value	Isolation-pos N=182	Isolation-neg No=110	*p* Value
Female sex	141 (48.3%)	45 (47.4%)	96 (51.3%)	0.6142	73 (51.8%)	68 (55.3%)	0.0545	85 (46.7%)	56 (50.9%)	0.5646
Underlying illness	108 (37.0%)	28 (25.9%)	80 (42.8%)	**0.0410**	61 (56.6%)	47 (38.2%)	0.8048	62 (57.4%)	42 (38.2%)	0.4767
Previous LB	57 (19.5%)	16 (28.1%)	41 (21.9%)	0.0772	42 (24.8%)	15 (12.2%)	**0.0109**	40 (22.0%)	17 (15.5%)	0.2266
Tick bite **	144 (49.3%)	41 (43.2%)	103 (55.1%)	0.3966	78 (46.2%)	66 (53.7%)	0.2510	93 (51.1%)	51 (46.4%)	0.5070
Multiple EM	40 (13.7%)	24 (25.3%)	16 (8.6%)	**0.0003**	28 (16.6%)	12 (9.8%)	0.1338	22 (12.1%)	18 (16.4%)	0.3931
Symptoms at the site of EM	138 (47.3%)	41 (43.2%)	97 (51.9%)	0.2086	66 (39.1%)	72 (58.5%)	**0.0015**	87 (47.8%)	51 (46.4%)	0.9064
Constitutional symptoms	99 (33.9%)	33 (34.7%)	66 (35.3%)	0.9686	59 (34.9%)	40 (32.5%)	0.7634	64 (35.2%)	35 (31.8%)	0.6471
IgM pos	95 (32.5%)	/	/	/	61 (36.1%)	34 (27.6%)	0.1628	40 (22.5%)	55 (50%)	**<0.0001**
IgG pos	169 (57.9%)	61 (64.2%)	108 (57.8%)	0.3590	/	/	/	105 (57.7%)	64 (58.2%)	0.9679
Skin culture pos	182 (62.3%)	60 (63.2%)	122 (65.2%)	0.8307	105 (62.1%)	77 (62.6%)	0.9679	/	/	/

pos, positive; neg *, negative or borderline; neg, negative; LB, Lyme borreliosis; EM, erythema migrans; boldface is used for *p*-values < 0.05; ** at the site of the EM skin lesion.

**Table 5 microorganisms-12-00185-t005:** Association between clinical parameters and the presence of *Borrelia* IgM antibodies in serum.

	B	Wald	df	Sig.
Intercept	2.256	9.290	1	0.002
Sex	0.154	0.253	1	0.615
Age	−0.040	11.549	1	**0.001**
Underlying chronic illness	0.037	0.012	1	0.914
Previous LB	0.109	0.077	1	0.781
Tick bite *	−0.123	0.160	1	0.689
Multiple EM	−1.183	7.212	1	**0.007**
Symptoms at the site of EM	0.098	0.311	1	0.752
Constitutional symptoms	0.002	0.329	1	0.995
Duration of EM at diagnosis	0.003	0.158	1	0.691
Diameter of EM at diagnosis	0.002	0.034	1	0.853
IgG-pos	−0.536	2.608	1	0.106
Skin culture-pos	−0.21	0.453	1	0.501

LB, Lyme borreliosis; EM, erythema migrans; pos, positive; boldface is used for *p*-values < 0.05; * at the site of the EM skin lesion.

**Table 6 microorganisms-12-00185-t006:** Association between clinical parameters and the presence of *Borrelia* IgG in serum.

	B	Wald	df	Sig.
Intercept	−0.670	0.890	1	0.345
Sex	−0.558	3.679	1	0.055
Age	0.029	6.508	1	**0.011**
Underlying chronic illness	−0.558	2.956	1	0.086
Previous LB	1.146	8.554	1	**0.003**
Tick bite *	0.076	0.064	1	0.800
Multiple EM	−0.725	2.285	1	0.131
Duration of EM at diagnosis	0.025	8.405	1	**0.004**
Diameter of EM at diagnosis	0.028	5.773	1	**0.016**
Symptoms at the site of EM	−0.680	5.344	1	0.021
Constitutional symptoms	−0.016	0.002	1	0.960
IgM-pos	−0.599	3.192	1	0.074
Skin culture-pos	0.390	1.626	1	0.202

LB, Lyme borreliosis; EM, erythema migrans; pos, positive; boldface is used for *p*-values < 0.05; * at the site of the EM skin lesion.

**Table 7 microorganisms-12-00185-t007:** Association between clinical parameters and the isolation of *Borrelia burgdorferi* sensu lato from skin.

	B	Wald	df	Sig.
Intercept	−0.633	0.919	1	0.338
Sex	−0.181	0.426	1	0.514
Age	0.018	2.820	1	0.093
Underlying chronic illness	−0.684	5.003	1	**0.025**
Previous LB	0.503	1.832	1	0.176
Tick bite *	0.260	0.864	1	0.353
Multiple EM	0.330	0.582	1	0.446
Symptoms at the site of EM	−0.105	0.136	1	0.712
Constitutional symptoms	0.242	0.622	1	0.430
Duration of EM at diagnosis	−0.005	0.532	1	0.466
Diameter of EM at diagnosis	0.003	0.069	1	0.793
IgM-pos	−0.228	0.527	1	0.468
IgG-pos	0.423	1.968	1	0.161

LB, Lyme borreliosis; EM, erythema migrans; pos, positive; boldface is used for *p*-values < 0.05; * at the site of the EM skin lesion.

## Data Availability

All the relevant data are included in the manuscript in an aggregated format.

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
