# Peer review of "Microbiologic Findings in a Cohort of Patients with Erythema Migrans"

_microorganisms, 2024, doi:10.3390/microorganisms12010185_

Round 1
Reviewer 1 Report
Comments and Suggestions for Authors
The manuscript reports findings from a well designed prospective study on patients with erythema migrans in Slovenia. The overall objective of the study is to correlate the presence of an EM lesion with the presence of spirochetes within lesion biopses and serologies (IgM and IgG). As noted by the authors, their findings align with numerous prior studies, most of which were conducted in the US, using similar approaches. The authors note that their serological approach uses updated (but not novel) antigens. While the work is well done, the studies are not particularly novel. The authors should include At the least, the authors should indicate that the studies were performed on patients in Slovenia, if only to distinguish it from similar studies performed in the North America.
Author Response
Thank you for reviewing our manuscript. As you noted, there are numerous studies about EM from North America but are rare from European countries. There was a lot of work to create this manuscript although it seems to be one of many on this subject. We believe it will be very useful for many physians as microbiologic analyses enable insights into etiology and dynamics of immune response in the course of typical EM.
As you suggested, we emphasized that study was performed on Slovenian patients; please look lines 107, 307 and 404.
Reviewer 2 Report
Comments and Suggestions for Authors
Keywords: erythema migrans, Lyme Borrelia, culture, serology
Is Lyme Borrelia the correct term ?
of Ixodidae 31
not cursive
(Sex, Age, Underlying chronic ill- 100
ness, Previous LB, Tick bite, Duration of EM at diagnosis, Diameter of EM at diagnosis, 101
Multiple EM, Symptoms at the site of EM, Constitutional symptoms),
Check orthography, initials should be small.
Pl give attention to the design of the tables and headlines
References: must be edited
improve language issues.
Comments on the Quality of English Languagesee report
Author Response
Thank you for reviewing our manuscript.
Regarding Keywords, it is really little bit strange “Lyme Borrelia” so we replaced these words with Lyme borreliosis and Borrelia afzelii.
As you suggested, we wrote Ixodidae not cursive (line 30), and initials of words in lines 99-101 in small letters.
We also checked and redesigned tables as much as possible.
We also edit reference chapter as it is requested by the journal. Thank you for your comment.
English was already editing by English speaking person in the field of microbiology.